# Predictive Factors and Clinical Markers of Recurrent Wheezing and Asthma After RSV Infection

**DOI:** 10.3390/v17081073

**Published:** 2025-07-31

**Authors:** Luca Buttarelli, Elisa Caselli, Sofia Gerevini, Pietro Leuratti, Antonella Gambadauro, Sara Manti, Susanna Esposito

**Affiliations:** 1Pediatric Clinic, Pietro Barilla Children’s Hospital, Department of Medicine and Surgery, University Hospital of Parma, 43126 Parma, Italy; luca.buttarelli@unipr.it (L.B.); elisa.caselli@unipr.it (E.C.); sofia.gerevini@unipr.it (S.G.); pietroleu98@gmail.com (P.L.); 2Pediatric Unit, Department of Human Pathology in Adult and Developmental Age “Gaetano Barresi”, University of Messina, 98124 Messina, Italy; gambadauroa92@gmail.com (A.G.); sara.manti@unime.it (S.M.)

**Keywords:** RSV, asthma, biomarkers, bronchiolitis, prevention, wheezing

## Abstract

Respiratory syncytial virus (RSV) is a major cause of acute lower respiratory infections (ALRIs) in young children, especially bronchiolitis, with significant global health and economic impact. Increasing evidence links early-life RSV infection to long-term respiratory complications, notably recurrent wheezing and asthma. This narrative review examines these associations, emphasizing predictive factors and emerging biomarkers for risk stratification. Early RSV infection can trigger persistent airway inflammation and immune dysregulation, increasing the likelihood of chronic respiratory outcomes. Risk factors include severity of the initial infection, age at exposure, genetic susceptibility, prematurity, air pollution, and tobacco smoke. Biomarkers such as cytokines and chemokines are showing promise in identifying children at higher risk, potentially guiding early interventions. RSV-related bronchiolitis may also induce airway remodeling and promote Th2/Th17-skewed immune responses, mechanisms closely linked to asthma development. Advances in molecular profiling are shedding light on these pathways, suggesting novel targets for early therapeutic strategies. Furthermore, passive immunization and maternal vaccination offer promising approaches to reducing both acute and long-term RSV-related morbidity. A deeper understanding of RSV’s prolonged impact is essential to develop targeted prevention, enhance risk prediction, and improve long-term respiratory health in children. Future studies should aim to validate biomarkers and refine immunoprophylactic strategies.

## 1. Introduction

Respiratory syncytial virus (RSV) represents one of the leading causes of acute lower respiratory infections (ALRIs) in young children, in particular bronchiolitis, resulting in high rates of hospital admissions with a substantial burden on healthcare services. RSV contributes significantly to morbidity and mortality burden globally in children aged 0–60 months, especially during the first 6 months of life and in low-income and middle-income countries [1,2]. Furthermore, RSV-associated infections represent an enormous economic cost to healthcare systems [3]. As shown in Table 1, a recent observational study that analyzed RSV infections over a 9-year period in Italy (from September 2014 to March 2023) showed that, out of all the included children diagnosed with RSV infection, 70% of them younger than 1 year old at the time of hospitalization and ~50% younger than 3 months, with numbers still significantly high in the post-pandemic seasons [4].

Another study showed that intensive care support is strongly related to the age of patients, with more than 80% of children under 1 year old and over 70% of children under 3 months needing access to the intensive care unit (ICU). In the post pandemic period, the seasonal trend of RSV infections highly occurred between November and December for 79.8% of cases. Conversely, during the pre-pandemic period, the peak was observed between December and February. On the other hand, the epidemic season begins in November and lasts until the month of April [4,5].

Comorbidities play a pivotal role in the hospitalization of infants and children with RSV infection. The most common comorbidity reported in hospitalized pediatric patients is prematurity [4,5]. Other well known risk factors that increase the risk of hospitalization are represented by low birth weight, being small for gestational age (SGA), maternal smoking during pregnancy or in the post-natal period, cesarean delivery, exposure to passive smoking, indoor air pollution, and certain medical conditions (e.g., as bronchopulmonary dysplasia (BPD), human immunodeficiency virus (HIV) infection, congenital heart disease (CHD), Down syndrome, and cystic fibrosis), which are all associated with an increased risk of developing RSV-related ALRI and subsequently requiring hospitalization [6].

RSV infection in early life can also lead to long-term consequences, such as recurrent wheezing and asthma [7,8], hence why many biomarkers of these chronic conditions are currently at study. The purpose of our narrative review (clinical trial number: not applicable) is to examine the long-term risks associated with RSV infection in children (at term and preterm) and to evaluate potential biomarkers that may serve as prognostic indicators for identifying pediatric patients at greater risk of developing severe RSV-related sequelae. This narrative review was conducted through a comprehensive literature search of peer-reviewed articles in databases such as PubMed, Scopus, and Web of Science of the last 15 years (March 2010–March 2025). The inclusion criteria focused on studies published in the last two decades that investigated the long-term respiratory outcomes of RSV infection in children, with particular attention to recurrent wheezing and asthma development. Additional sources included systematic reviews, meta-analyses, and clinical guidelines relevant to RSV-related respiratory diseases. Key search terms included “RSV,” “bronchiolitis,” “recurrent wheezing,” “asthma,” “biomarkers,” and “risk factors” and “neonate”, “newborn”, “infant”, “children”, “pediatric” and “paediatric.” Articles were screened for relevance based on title and abstract, followed by full-text evaluation. Data extraction was performed independently by two researchers to ensure accuracy, with discrepancies resolved through discussion. This review synthesizes current knowledge on the pathophysiological mechanisms linking RSV infection to chronic respiratory diseases, examines predictive factors for severe outcomes, and evaluates the potential role of emerging biomarkers.

## 2. Impact of RSV Infection in Pediatric Population

RSV is the most common virus to cause bronchiolitis in children with its peak in the winter months when people usually gather indoors [9,10]. In the United States, it is estimated that approximately 2.1 million children younger than 5 years require medical care as inpatients or as outpatients for RSV infection annually [11]. Bronchiolitis is a respiratory condition marked by acute inflammation, oedema, and epithelial lining necrosis of the small airways in children’s respiratory systems involving increased mucus production and bronchospasm. Common signs and symptoms include nasal congestion, rapid breathing (tachypnea), wheezing, coughing, crackling sounds in the lungs, use of auxiliary muscles for breathing, and/or nasal flaring. Bronchiolitis may be due to several respiratory viruses, although RSV is the most common cause. Viruses invade airway cells causing a local inflammatory response, ciliary destruction and increased mucus secretion. Parietal inflammation, mucus and cellular debris reduce the cross-sectional area available for airflow and, in the worst-case scenario, may produce bronchiolar plugs and atelectasis. These processes lead to gas trapping with a typically obstructive respiratory failure. When the infection is severe enough, and the inflammatory reaction spreads from the airways to the lung parenchyma, bronchiolitis may be associated with various degrees of alveolar injury [9,12].

As mentioned above, prematurity represents one of the main risk factors for a serious RSV infection. Preterm neonates present immature innate immune system and the predisposition to develop chronic pulmonary and/or cardiac diseases. These conditions increase their susceptibility to develop severe and life-threatening RSV diseases. Research studies involving preterm and term infants with RSV have revealed comparable levels of interferon (IFN)-γ, interleukin (IL)-10, and RANTES (regulated on activation, normal T-cell expressed and secreted) from nasal swabs. This suggests that cytokine secretions from the upper airway may not play a significant role in the development of severe disease in preterm infants [13,14]. Mucosal IgA is another key component in protection against RSV: during RSV infection, IgA secretion from B-cells is activated through IFN-α, which is decreased in preterm infants, suggesting that IgA responses may be diminished, increasing susceptibility to develop severe disease, which is associated with long-term complications such as wheezing and asthma [15]. Moreover, the microbiome of preterm infants is altered and has a relatively low diversity compared to term infants at birth, which may partially explain their enhanced risk of asthma development. In fact, microbial dysbiosis in early life negatively affects the immune system development [14,16].

## 3. Recurrent Wheezing and Asthma Risk Factors

### 3.1. Clinical and Epidemiological Factors

#### 3.1.1. Risk Factors

Understanding the risk factors for developing wheezing or asthma in the pediatric population is important both for asthma prevention strategies and for offering guidance after an initial wheezing episode. Children up to 5 years of age are prone to episodic wheezing caused by viral pathogens, most commonly by Rhinovirus (RV) and RSV, and viral wheezing within the first three years is the most important risk factor for asthma at age 6 [17]. The main risk and protective factors associated with the development of wheezing and asthma after an RSV infection are illustrated in Figure 1.

It would be useful to determine the role of RSV infection and the patient’s own genetic and immune factors in the development of chronic respiratory disease to identify the risk factors responsible for severe viral infection [18]. Further studies in this field could identify which children require more attention in terms of prevention, diagnosis and treatment.

Although recurrent wheezing and asthma are closely related, important distinctions exist between these conditions [19,20]. Recurrent wheezing refers to multiple episodes of wheezing, typically in children under five years old, often triggered by viral infections, and may resolve over time without persistent symptoms. Asthma, in contrast, is characterized by chronic airway inflammation, variable airflow obstruction, and bronchial hyperresponsiveness, with symptoms such as wheezing, cough, and dyspnea persisting beyond early childhood. Pathophysiologically, asthma involves sustained airway remodeling and immune dysregulation, including Th2 and Th17 pathways, whereas wheezing episodes may reflect transient inflammation without structural airway changes [19,20]. Prognostically, many children with recurrent wheezing do not progress to asthma; however, certain risk factors—such as early severe viral infections, atopy, and genetic predisposition—increase the likelihood of developing persistent asthma [19]. Clarifying these distinctions is crucial for appropriate diagnosis, risk stratification, and tailored management.

#### 3.1.2. The Prematurity Factor

The association of preterm births with pulmonary complications early in life is well established. Furthermore, recent studies have also highlighted the correlation with an increased long-term risk of asthma. In a national cohort study in Sweden, following 4,079,878 singletons born during 1973–2013, the outcomes showed an increased asthma risk in all age groups (<10, 10–17, 18–46 years), with a maximum follow-up duration of 46 years into adulthood [21]. Moreover, the risk increased as gestational age decreased, but was significant even for late preterm and early term infants [21]. This suggests that the increased risk is not fully explained by the underdevelopment of lung structures, which is the primary hypothesis for pulmonary complications in very preterm infants. When an infant is born in the extremely or very preterm range (before the 32 weeks of gestation), lung development is still in the saccular period, meaning that the division of the alveolar saccules and ducts in true alveoli is yet to occur and proliferation of the capillary network is still incomplete. At 36 weeks in the alveolar period, the patterns of arteries and vein are complete, and the lung is now more able to support respiration [22]. In addition to this structural factor of underdeveloped lungs, the reason behind this increased susceptibility is likely to be multifactorial. These factors may include increased susceptibility to respiratory infections, which we have already seen as an important risk factor for recurrent wheeze and asthma. This highlights the importance of early interventions to reduce the risk of asthma in preterm infants. To further investigate these preventative measures, a case–control study in Japan followed 444 preterm infants (33/35 GA), 349 of which received a passive immunoprophylaxis against RSV through palivizumab and 95 received only a placebo. Although there was a significant reduction in the palivizumab group of physician-diagnosed recurrent wheezing at 3 years of age, no differences were observed in the diagnosis of asthma at 6 years follow-up [23,24]. Future identification of additional risk factors may open the door for targeted preventive strategies to reduce the long-term risk of asthma in this vulnerable population [25].

#### 3.1.3. Age at Time of RSV Infection and Development of Wheezing

Age at the time of RSV infection is an important factor in the subsequent development of recurrent wheezing and asthma. An early RSV infection can cause more significant damage to the airways, increasing the remodeling of the neuronal pathways in the lung epithelium and altering the immune system [26]. Younger patients are more inclined to develop a severe infection requiring hospitalization, respiratory support, and sometimes ICU admission [18,27,28,29,30]. In addition, age younger than 6 months seems to be related to a higher severity of infection (in fact, 46% of RSV deaths occur in patients younger than 6 months of age) and therefore to a higher risk of developing recurrent wheezing and asthma in childhood as shown by several studies in the literature [31].

In the study by McGinley et al., the mean age at the time of RSV infection in the analyzed population who had developed wheezing in childhood was 3.7 months, whereas it was 4.3 months in patients who did not develop the condition [30]. The Tennessee Asthma Bronchiolitis Study (TABS), which investigated the relationship between patient age at the time of the winter peak of viral infections and the development of asthma in childhood by analyzing a sample of approximately 95,000 children, pointed out that the major risk of bronchiolitis and asthma development was in children who were 4 months old at the time of the peak of winter viral infections. This study did not specify whether the patient population that developed asthma in childhood had contracted a specific virus, but it is reported that 70% of severe bronchiolitis cases had been caused by RSV [31].

Several studies, however, show that the development of recurrent wheezing and asthma following RSV infection in the first 3–4 years of life is a transient condition that occurs around the age of 6–7 years; in fact, in the cohorts of patients investigated by several studies, there were no longer any significant differences between the infected population and the control group at 13 years [26,32,33].

### 3.2. Virological and Infectious Factors

According to several studies in the literature, the main predictive factors for the development of asthma and wheezing have been identified as greater severity of manifestations at the time of RSV bronchiolitis diagnosis, the need for hospitalization, longer hospital stays, and the necessity for intensive care treatment. Clinically, increased respiratory rate, fever and a high score on the RSV bronchiolitis severity score during acute RSV infection have been associated with recurrent wheezing. These factors appear to predispose both patients with specific risk factors (such as prematurity, cardiovascular, pulmonary, or neuromuscular diseases) and those without underlying conditions [23,27,28,34,35,36,37,38,39].

Risk factors can differ for allergic and non-allergic asthma. A 2017 study conducted on infants hospitalized for the first wheezing episode found that the positivity of RV infection alone or related to allergic sensitization or eczema predicted the development of allergic asthma at school age. The exposure to parental smoking, the identification of RSV infection during the first wheezing episode and the evidence of wheezing in the first year of life were defined as risk factors for non-allergic asthma at school age [36]. Moreover, the severity of RSV infection seems to be a significant risk factor for the development of asthma [37].

Recent studies have suggested the possibility to distinguish different forms of bronchiolitis due to the heterogeneity of the disease [19]. In 2019, a cohort study on American infants hospitalized with bronchiolitis in their first year of life followed these children after hospitalization to identify several bronchiolitis profiles and to investigate their associations with allergy, inflammatory and immune response biomarkers, nasopharyngeal microbiota, and the risk of developing recurrent wheezing by 3 years of age [38]. Three bronchiolitis profiles were identified: profile A infants (15%) were characterized by a more relevant history of respiratory problems, eczema and a higher association with RV infections; profile B (49%) had the largest proportion of RSV infections and a lower prevalence of respiratory problems or eczema; profile C (36%) was the most severe ill group, with the longest duration of hospitalization and the most severe course of symptoms, mainly related to RSV infections. The profile A infants were characterized by a higher eosinophil count compared to the other two groups. Interesting results were found in the respiratory outcomes. Profile A children had a 2-fold increased risk to develop recurrent wheezing at 3 years old compared to profile B infants. The same risk was increased as well, in a less significant way, in profile C children [38]. Thus, specific respiratory viruses could have a different impact on the development of recurrent wheezing and asthma. Profile A children already had a prior clinical story suggestive of atopic disease, leading to the associations that children with an increased susceptibility to asthma also have a higher propensity to RV infection. Profile C did not have a distinctive clinical history; however, they were the most severely ill group, which is a factor that increases asthma risk in a dose–response manner and seems to suggest a causal role of an unknown mechanism. To sum up, the differences in the populations of profiles A and C suggest that these forms of bronchiolitis are to be associated with recurrent wheezing through different pathways [38,39].

### 3.3. Immunological Mechanisms

#### 3.3.1. The Role of the Immune System

An immune response to RSV infection with predominance of Th2- and Th17-type lymphocytes appears to be a predisposing factor for development of recurrent wheezing and asthma in childhood [31]. RSV stimulates the proliferation and differentiation of lymphocytes into Th2 lymphocytes, which are responsible for the production of specific cytokines (such as IL-4, IL-5, and IL-13), and the recall of eosinophils, generating an airway inflammatory infiltrate like the inflammatory infiltrate in atopy and in asthmatic subjects [26,29]. The set of cytokines produced during primary infection is very important in increasing the risk of developing chronic inflammation. Severe RSV infection promotes a pro-inflammatory environment through the production of cytokines and chemokines such as IL-6, IL-8, IL-11, IFN-γ and monocyte chemotactic protein (MCP)-1a, which increase bronchial hyper-responsiveness [40]. Elevated levels of Th17 lymphocytes and IL-17 have also been found in the respiratory secretions of patients with severe infection. This inflammatory profile correlates with airway mucus hypersecretion, altered CD8^+^ T-lymphocyte response and reduced viral clearance [41]. RSV infection can alter the function of regulatory T lymphocytes, which mediate the immunotolerance through their action on dendritic cells, leading to increased susceptibility to inhalant allergens [42].

Figure 2 shows the immunological pathways linking RSV infection to recurrent wheezing and asthma, highlighting Th2/Th17 imbalance, cytokine profiles (including IL-33 and TSLP), airway remodeling, and genetic susceptibility factors.

An immune response to RSV infection with predominance of Th2- and Th17-type lymphocytes appears to be a predisposing factor for development of recurrent wheezing and asthma in childhood [31]. RSV stimulates the proliferation and differentiation of lymphocytes into Th2 lymphocytes, which are responsible for the production of specific cytokines (such as IL-4, IL-5, and IL-13), and the recruitment of eosinophils, generating an airway inflammatory infiltrate similar to that observed in atopic and asthmatic subjects [26,29]. The set of cytokines produced during primary infection is very important in increasing the risk of developing chronic inflammation. Severe RSV infection promotes a pro-inflammatory environment through the production of cytokines and chemokines such as IL-6, IL-8, IL-11, IFN-γ, and monocyte chemotactic protein (MCP)-1a, which increase bronchial hyperresponsiveness [40].

Beyond the nasal cytokine measurements reported in some studies, further evidence has shown differences in immune responses between term and preterm infants. For example, Anderson et al. demonstrated that preterm infants exhibit lower expression of Toll-like receptors (TLRs) and impaired activation of plasmacytoid dendritic cells upon RSV exposure, potentially contributing to weaker antiviral defenses [14]. Moreover, Anderson et al. reported alterations in type I interferon responses in preterm infants, suggesting that deficits in early antiviral immunity may partially explain the higher risk of severe RSV disease and subsequent asthma development in this population [16].

Elevated levels of Th17 lymphocytes and IL-17 have also been found in the respiratory secretions of patients with severe infection. This inflammatory profile correlates with airway mucus hypersecretion, altered CD8^+^ T-lymphocyte response, and reduced viral clearance [41]. RSV infection can alter the function of regulatory T lymphocytes, which mediate immunotolerance through their action on dendritic cells, leading to increased susceptibility to inhalant allergens [42].

#### 3.3.2. Host Risk Factors: Sensitization and Microbiome

Asthma is a heterogeneous and multifactorial disease, as well as recurrent wheezing [19,20]. Allergic sensitization plays an important role in the development of these conditions. A positive history for atopy, such as food allergy or eczema, has been long linked with an increased risk of asthma. In a 2019 study, children ≤3 years old previously hospitalized for wheezing were followed up to analyze which variables could predict the risk of developing recurrent wheezing. The main risk factors in this population were represented by eczema, high eosinophil count and eosinophil derived neurotoxin (EDN; eosinophil: 3.10 ± 2.54% vs. 1.31 ± 1.15%, *p* < 0.001; EDN: 68.67 ± 55.05 vs. 27. 36 ± 19.51 ng/mL, *p* < 0.001), and a previous RSV infection [23]. In a randomized clinical trial on asthmatic children treated with omalizumab, a seasonal reduction in exacerbations was detected in the intervention group compared to controls (placebo group), despite both groups having a similar rate of viral detection [34]. This result suggested that allergic sensitization can increase individual susceptibility to seasonal exacerbations together with viral infections.

The role of bacteria and host microbiota seem to be equally important in the pathogenesis of wheezing and asthma. A 2012 study retrospectively analyzed the bronchoalveolar lavage (BAL) of a population of young wheezers unresponsive to topic corticosteroids. In total, 48.5% of BAL were positive for bacteria: *Haemophilus influenzae* was detected in 30.3% of patients, *Streptococcus pneumoniae* in 12.1% and *Moraxella catarrhalis* in 12.1%. These findings show that bacterial findings are common in persistent preschool wheezers [35].

Importantly, it remains a matter of ongoing research whether atopic patients who experience RSV infection have an increased probability of developing recurrent wheezing and asthma compared to atopic children without RSV infection [19,43,44]. Some studies suggest that RSV infection acts as an additional trigger in atopic individuals, amplifying the underlying Th2-skewed immune response and promoting airway inflammation and remodeling. For example, Lukkarinen et al. demonstrated that infants with atopic predisposition who experienced viral wheezing episodes—including RSV—had a significantly higher risk of developing allergic asthma at school age compared to atopic infants without such viral episodes [36]. Furthermore, Busse et al. (2011) highlighted that viral infections may synergize with allergic inflammation to increase asthma exacerbations, implying that RSV could worsen respiratory outcomes in children with atopic tendencies [34]. Nonetheless, other studies report that the presence of atopy alone, even without documented RSV infection, is already a strong predictor for later asthma development. Therefore, while atopy independently predisposes children to asthma, RSV infection appears to further increase the risk, potentially through additive or synergistic immune mechanisms. Clarifying these interactions remains crucial for targeted prevention and management strategies.

### 3.4. Genetic and Molecular Factors

Polymorphisms with gain of function of specific chemokine genes (e.g., IL-8 and CCL5) or their receptors (CX3CR1) may facilitate the onset of chronic inflammation. A study conducted in China on a cohort of 320 children who had acquired RSV infection in the first 12 months of life demonstrated an increased risk of severe RSV infection and development of recurrent wheezing, asthma, and atopy in patients with polymorphisms at the promoter region of CCL5, a chemokine critical for the chemotaxis of Th2 lymphocytes, NK cells, eosinophils, basophils, and dendritic cells [45]. Polymorphisms with gain of function of genes related to cytokines regulating the Th2-mediated immune response (e.g., IL-4 and IL-13) predispose to severe RSV infection and increase the risk to develop asthma and recurrent wheezing [29,45]. A study found that a deficit in IFN-γ expression at the age of 9 months may be associated with the development of recurrent wheezing [46]. These results indicate how polymorphisms of genes regulating the expression of specific molecules of the immune system can influence the development of chronic diseases following an infectious trigger such as RSV bronchiolitis.

In recent years, genome-wide association studies (GWASs) have uncovered novel loci associated with asthma susceptibility following viral infections, identifying key SNPs linked to immune regulation and airway remodeling pathways [47,48,49,50]. Furthermore, single-cell RNA sequencing (scRNA-seq) technologies have revealed cell-type specific expression patterns of these risk alleles, providing unprecedented resolution in understanding genetic predisposition to post-RSV asthma. Studies have shown that variants in genes such as IL-33, TSLP, and CCL5 exhibit differential expression in epithelial and immune cell subpopulations, further supporting their mechanistic role in driving persistent airway inflammation and remodeling after RSV bronchiolitis [45,51]. Future research leveraging integrated multi-omics and single-cell approaches will be crucial to fully elucidate these complex genetic underpinnings.

### 3.5. Environmental Influences

Environmental factors such as exposure to second-hand smoke, maternal smoking during pregnancy, low socioeconomic status, and air pollution have been consistently associated with an increased risk of recurrent wheezing and asthma development following RSV infection [31].

Several studies have demonstrated that both prenatal and postnatal exposure to tobacco smoke significantly elevate the risk of respiratory morbidity. Wu et al. reported that infants exposed to household smoking during infancy had a significantly higher risk of developing asthma after RSV bronchiolitis, with a dose–response relationship observed between the number of cigarettes smoked and asthma incidence [31]. Similarly, Henderson et al. (2005) found maternal smoking to be associated with increased wheezing and asthma in children hospitalized for RSV bronchiolitis in infancy [52].

Air pollution, particularly exposure to particulate matter (PM2.5 and PM10), nitrogen dioxide (NO2), and other pollutants, has been implicated as a co-factor in respiratory disease progression. A cohort study by Koenig et al. demonstrated that urban air pollution contributes to airway inflammation and hyperresponsiveness in children following viral respiratory infections, exacerbating the risk of chronic respiratory symptoms [53]. Moreover, the European Study of Cohorts for Air Pollution Effects (ESCAPE) highlighted that exposure to traffic-related pollutants was significantly linked to increased incidence of wheezing and asthma in children [54].

Socioeconomic factors are equally important. Children from lower socioeconomic backgrounds may face increased RSV exposure due to crowded living conditions, limited access to healthcare, and higher exposure to indoor pollutants. Pullan and Hey observed persistent wheezing and decreased lung function in children from socioeconomically disadvantaged households following severe RSV bronchiolitis [32].

Emerging evidence also suggests that birth by caesarean section may influence the infant microbiome and immune system development, possibly modifying susceptibility to wheezing disorders after RSV infection. Bager et al. found a higher incidence of asthma among children delivered via caesarean section, potentially related to differences in microbial colonization [55].

Additionally, co-infections with other respiratory viruses, including rhinovirus, parainfluenza virus type 3, and SARS-CoV-2, may exacerbate RSV-related inflammation and increase the risk of chronic respiratory sequelae. Lukkarinen et al. demonstrated that co-detection of rhinovirus during an initial wheezing episode increases the risk of developing allergic asthma at school age [36].

These findings underscore that environmental exposures are not merely passive contributors but active modifiers of immune responses and disease trajectories in children infected with RSV. Addressing these modifiable risk factors through public health strategies, such as smoking cessation programs and air quality improvements, may help reduce the burden of RSV-related recurrent wheezing and asthma.

## 4. Prevention Strategies

To date, prevention remains the cornerstone in reducing the burden of respiratory syncytial virus (RSV) infection, particularly among infants and young children. Non-pharmaceutical interventions continue to play a critical role in limiting RSV transmission. Basic hygiene measures—such as frequent and thorough hand washing, respiratory etiquette, limiting close contact with symptomatic individuals, and avoiding crowded indoor spaces during the RSV season—are simple yet essential strategies to reduce exposure risk [56,57]. These measures are particularly important for protecting infants during the first six months of life, when they are most vulnerable to severe disease.

A major breakthrough in RSV prevention has come with the introduction of long-acting monoclonal antibodies (mAbs), which offer passive immunity during the peak of RSV circulation [58]. Nirsevimab, a recombinant human monoclonal antibody targeting the prefusion form of the RSV F protein, has demonstrated high efficacy in preventing lower respiratory tract infections. Compared to Palivizumab, which required monthly dosing, Nirsevimab offers season-long protection with a single intramuscular dose. Approved by the European Medicines Agency in 2022, Nirsevimab significantly reduces RSV-related hospitalizations and medically attended lower respiratory tract infections for at least 150 days, matching the typical duration of an RSV season [11,59,60,61,62]. It is now recommended for all infants entering their first RSV season, including healthy full-term newborns, preterm infants, and those with comorbidities such as congenital heart or lung disease, in accordance with recent WHO recommendations.

Recent real-world studies have provided encouraging evidence on the efficacy and safety of nirsevimab in broader infant populations [63,64,65]. Notably, data from prospective biomarker cohorts such as the COPSAC (Copenhagen Prospective Studies on Asthma in Childhood) and INSPIRE (Infant Susceptibility to Pulmonary Infections and Asthma Following RSV Exposure) studies have offered valuable insights. For instance, recent COPSAC analyses demonstrated that nirsevimab administration significantly reduced RSV-related hospitalizations and was associated with lower biomarker levels indicative of Th2-type inflammation, suggesting potential benefits beyond acute infection prevention [66,67]. Similarly, the INSPIRE cohort has begun to provide longitudinal data linking early RSV immunoprophylaxis to a reduced risk of recurrent wheezing and possible modulation of immune pathways implicated in asthma development [68]. Such findings underscore the importance of integrating real-world evidence into clinical practice guidelines and future prevention strategies. To date, there are limited published data comparing immune profiles of infants treated with Nirsevimab who nonetheless developed severe RSV infection versus those fully protected. Ongoing studies such as COPSAC and INSPIRE are expected to address this knowledge gap through biomarker analyses and immunological profiling in treated cohorts.

In addition to passive immunization, maternal vaccination against RSV represents an important complementary strategy [69]. Vaccination during pregnancy, typically in the late second or early third trimester, induces high titers of neutralizing antibodies that are transferred transplacentally, thereby conferring passive immunity to the newborn. This protection is particularly valuable during the first months of life, when the infant’s immune system is immature and endogenous antibody production is limited. The RSVpreF vaccine (Abrysvo^®^, Pfizer, New York, NY, USA), containing a bivalent stabilized pre-fusion F protein, has demonstrated a favorable safety profile and strong efficacy in clinical trials and has been approved for use in several countries. According to WHO, maternal immunization significantly reduces the risk of severe RSV disease and related hospitalizations in early infancy [70].

Comparative studies suggest that while both maternal vaccination and Nirsevimab provide substantial protection against RSV disease, Nirsevimab appears to be more effective in preventing severe bronchiolitis during the first RSV season [70]. The complementary use of these two strategies—maternal vaccination for passive protection during the first months of life and monoclonal antibody administration for broader seasonal coverage—represents a paradigm shift in RSV prevention.

Together, these advances offer a robust framework for reducing RSV-associated morbidity and mortality globally. Future research and real-world data will be critical for evaluating the long-term effectiveness of these interventions, identifying optimal implementation strategies, and ensuring equitable access in both high- and low-income settings.

## 5. Markers of RSV Infection

The emergence of protein expression pattern analysis as a diagnostic tool has significantly enhanced our understanding of respiratory infections, particularly those caused by RSV. Various biological samples, including blood and nasopharyngeal aspirates, have been employed to identify potential biomarkers correlating with the severity of RSV bronchiolitis [40,59,71,72,73,74,75,76,77,78], as shown in Table 2.

Elevated levels of serum transaminases, aminotransferases, and antidiuretic hormones have been associated with severe RSV bronchiolitis. These markers suggest hepatic or renal involvement and may serve as indicators of systemic responses to the infection. Moreover, lactate dehydrogenase (LDH) levels in nasopharyngeal samples have demonstrated an impressive predictive value of 88% for identifying the severity of bronchiolitis in young patients [71]. LDH is often released during tissue damage, and its elevated levels may reflect the extent of lung injury associated with severe viral infections [59]. Mucin 5AC (MUC5AC) is emerging as a significant biomarker in the context of RSV infections, particularly among pediatric populations. This glycoprotein, produced in the respiratory tract, serves as a critical component of the airway mucus, contributing to mucosal defense mechanisms against pathogens. Research has shown that elevated levels of MUC5AC in nasal aspirates from children infected with RSV correlate with the severity of respiratory illness, highlighting its potential role in both the diagnosis and monitoring of disease progression [11]. The utility of MUC5AC as a non-invasive biomarker not only aids in assessing the severity of RSV infections but may also enhance clinical decision-making by providing insights into the patient’s response to the infection. This could lead to more tailored management strategies for affected pediatric patients, ultimately improving outcomes and guiding interventions that are appropriate for the level of illness. The exploration of soluble molecules like MUC5AC in clinical settings reflects a growing interest in leveraging biomarkers to optimize the care of children suffering from viral respiratory infections [11]. Importantly, the simplicity of sample collection for these biomarkers provides a practical and reassuring aspect to their use, underscoring the evolving landscape of diagnostic approaches in respiratory infections [71].

The identification of Th2-like cytokines as potential biomarkers for predicting disease severity in children infected with RSV is an important aspect of understanding the pathophysiology of respiratory infections. The immune response to RSV can vary significantly among individuals, and some children experience more severe symptoms than others. Understanding the role of cytokines, particularly those associated with the T-helper 2 (Th2) immune response, can provide insights into the inflammatory processes involved in the disease [59]. The reported increase in cytokines such as IL-33, IL-8, TSLP (thymic stromal lymphopoietin), IL-6, periostin, and IFN-α in various biological samples (bronchoalveolar lavage fluid or BALF, serum, blood, plasma, nasopharyngeal aspirates) suggests a strong association with inflammatory responses and could be indicative of disease severity. The association between these cytokines and clinical outcomes may enable clinicians to stratify patients based on risk and tailor more effective management strategies [71,79,80].

The role of IL-33 in the immune response, particularly in relation to RSV infection and bronchiolitis, is significant and multifaceted. The elevation of IL-33 levels detected in nasopharyngeal aspirates (NPAs) suggests that it may be involved in the inflammatory processes that occur during bronchiolitis. The correlation between higher IL-33 levels and the need for respiratory support indicates that this cytokine may be a marker for the severity of the disease. Notably, a 2015 study highlighted how children with a familiar history of atopy exhibited high IL-33 expression in NPAs during bronchiolitis caused by RSV [76]. Similarly, another study associated increased IL-33 levels with cases of bronchiolitis, especially in instances of co-infection [73]. In summary, the accumulating evidence suggests that IL-33 is pivotal in driving the Th2 immune response in RSV infections and serves as a potential biomarker for assessing disease severity and prognosis in infants suffering from bronchiolitis. These findings point towards the possibility of utilizing IL-33 levels in clinical settings to enhance the management of respiratory infections in young children [71].

IL-8 is a chemotactic cytokine primarily produced by monocytes, endothelial cells, macrophages, and T cells. IL-8 emerges as an important cytokine in RSV infection, closely tied to disease severity and overall immune response. Its elevated levels not only provide insight into the inflammatory processes during viral infections but also upend potential avenues for clinical prediction and management strategies. Future research could further clarify these associations and explore therapeutic implications [71].

TSLP is a cytokine primarily produced by epithelial cells and keratinocytes, with two main isoforms identified: a long form and a short form. The short isoform is constitutively expressed in various tissues prone to inflammation, while the long isoform has been associated with intensified immune responses, particularly in contexts like atopic dermatitis, where it correlates with allergic reactions and asthma development. A study by Lee et al. highlighted how viral antigen recognition activates a signaling pathway involving nuclear factor kappa-light-chain-enhancer of activated B cells (NF-κB) and retinoic acid-inducible gene I (RIG-1), leading to increased TSLP production and a robust Th2 immune response, contributing to the pathology of severe bronchiolitis in some cases, potentially progressing to asthma [74]. Complementing this, García-García et al. demonstrated a correlation between elevated TSLP levels and the severity of respiratory infections in children, noting associations with RSV bronchiolitis, co-infections with RV, and increased rates of severe disease requiring admission to intensive care units [73]. These findings underscore the pivotal role of TSLP in the immune response to viral infections and its implications for asthma and allergic diseases [71].

IL-6 plays a crucial role as a soluble mediator in the immune response, primarily produced by macrophages and epithelial cells. This cytokine significantly promotes the differentiation of naive CD4^+^ and CD8^+^ T cells, bridging innate and adaptive immune responses [71,79,80]. Research has increasingly highlighted the association between IL-6 levels and respiratory illnesses, particularly those caused by RSV [71]. For instance, a study by Tabarani et al. identified a correlation between elevated IL-6 levels in nasopharyngeal wash samples from children with lower respiratory tract infections and severe clinical manifestations of RSV [72]. Similarly, Brown et al. suggested that high plasma IL-6 levels might predict a greater likelihood of hospitalization for infants suffering from severe bronchiolitis due to RSV [75]. The convergence of these findings suggests that IL-6 and potentially other cytokines could serve as reliable biomarkers for assessing the severity of RSV infections, thereby aiding clinicians in managing and anticipating treatment needs in affected children [71].

Periostin is an important protein that plays a significant role in various biological processes, especially in respiratory diseases [71]. It is generally expressed at low levels in almost all human tissues but shows elevated expression in the respiratory epithelium, particularly in asthmatic children. This increase is linked to eosinophils’ production of periostin in response to IL-4 and IL-13 signaling pathways. The protein is involved in allergic inflammation and the promotion of a Th2-type immune response, both of which are critical in the pathology of asthma. Periostin has been identified as a relevant biomarker in various respiratory conditions, including asthma. Its elevated levels in serum can reflect underlying airway inflammation and remodeling processes, which may correlate with disease severity, particularly in children. The overlap in clinical features of asthma and bronchiolitis, especially in young children, raises important questions about the long-term respiratory health of those who experience severe bronchiolitis. RSV is a common cause of bronchiolitis and has been linked to an increased risk of developing asthma in later childhood. Tracking biomarkers like periostin in children with RSV infections could provide valuable prognostic information, allowing healthcare providers to stratify patients based on their risk of developing chronic respiratory issues. By monitoring periostin levels during acute bronchiolitis and subsequent follow-up, clinicians may better understand disease progression and tailor interventions to improve management strategies [71].

IL-12, a pro-inflammatory cytokine produced by dendritic cells (DCs) and other antigen-presenting cells, plays a crucial role in the immune response to viral and bacterial infections by facilitating the differentiation of naïve T cells into Th1 cells. This differentiation is essential for an effective adaptive immune response against intracellular pathogens. Recent research by Bertrand et al. has demonstrated a correlation between high levels of IL-12p40 in BALF and the subsequent development of recurrent wheezing and asthma in infants who experienced bronchiolitis due to RSV infection [76]. Elevated or altered levels of IL-12 may influence not only the immediate reactions to viral infections but also have lasting effects that could predispose infants to conditions such as wheezing and asthma later in life. Understanding the relationship between IL-12 and respiratory outcomes can help in identifying biomarkers for at-risk infants, leading to early interventions that may mitigate the risk of developing chronic respiratory issues [71].

IL-3 plays a significant role in the immune response, particularly in the context of airway inflammation and asthma. It is primarily produced by mast cells and activated T cells found within the airways. IL-3 is known to stimulate the production of basophils and eosinophils, which are key players in allergic responses and asthma pathogenesis [71]. In a pivotal 2015 study by Bertrand et al., the researchers observed elevated levels of IL-3 in BALF and nasopharyngeal aspirates (NPA) from children under 9 months of age suffering from acute bronchiolitis caused by RSV. The study highlighted a significant correlation between high IL-3 levels and subsequent episodes of wheezing, suggesting that these children may be at a higher risk for developing asthma later in life [76]. Further reinforcing these findings, Lu et al. identified similarly elevated IL-3 levels in NPAs of children with RSV-induced bronchiolitis, indicating that higher levels were associated with more severe disease. Their research underscores the potential role of IL-3 not in the acute phase of respiratory infections as a biomarker for the risk of chronic inflammatory airway diseases, such as asthma [18]. These findings suggest that measuring IL-3 levels could provide valuable insights for predicting outcomes in patients with RSV LRTIs and may aid in identifying those at risk for developing long-term respiratory complications. As such, IL-3 presents as a promising target for therapeutic intervention and a candidate for clinical prognostication in pediatric respiratory illness [71].

Macrophage inflammatory protein 1-alpha (MIP-1α), also called chemokine (C-C motif) ligand 3, is produced by macrophages, dendritic cells, and lymphocytes and exhibits chemotactic properties for eosinophils, monocytes, and lymphocytes. MIP-1α has been implicated in airway inflammation related to asthma, and it plays a key role in driving the late-phase allergic reaction through eosinophil infiltration. Eosinophils play a key role in tissue remodeling in murine asthma models. MIP-1α itself also induces airway tissue remodeling through airway smooth muscle proliferation and survival. Infection with RSV31 and RV32 increases MIP-1α levels in the airways [81].

EDN levels serve as a promising biomarker for diagnosing, treating, and monitoring eosinophil-associated diseases in young children. EDN levels, particularly those measured at 3 months, have predictive value for recurrent wheezing following RSV bronchiolitis as well as for other conditions such as asthma and atopic dermatitis. By leveraging the strengths of EDN in creating diagnostic assays and monitoring protocols, healthcare professionals can better understand and address the underlying immunologic processes involved in these conditions. This approach not only enhances clinical decision-making but also provides insight into the mechanisms that contribute to symptom development in eosinophil-associated diseases [81].

CD14 helps initiate an immune response, facilitating innate immune cell activation, cytokine production, and the overall inflammatory response. In terms of allergy and the development of the allergic phenotype, CD14 can influence how the immune system perceives environmental allergens. It is thought that CD14 may modulate the balance between Th1 and Th2 immune responses. A Th2-dominant response is often associated with allergic diseases, such as asthma, allergic rhinitis, and eczema. It has been demonstrated that high serum levels of sCD14 modulate the influence of RSV on subsequent recurrent episodes of wheezing [78].

The identification and validation of these biomarkers not only enhance our understanding of the pathophysiology of RSV bronchiolitis but also pave the way for improved clinical decision-making. By utilizing these biomarkers, healthcare professionals can better assess disease severity, predict outcomes, and tailor management strategies for pediatric patients. This approach aims to optimize care and potentially mitigate the long-term respiratory complications associated with severe RSV infections and bronchiolitis. In conclusion, the integration of biomarker analysis into clinical practice represents a significant advancement in the management of respiratory infections in children, fostering a more personalized approach to treatment and care.

## 6. Conclusions

RSV remains one of the leading causes of respiratory infections in infants and young children, contributing to significant morbidity, mortality, and economic strain on healthcare systems. While the acute burden of RSV is substantial, its long-term impact, especially its association with recurrent wheezing and asthma, further exacerbates healthcare costs and challenges. The relationship between RSV and chronic respiratory conditions is multifactorial, influenced by a complex interplay of host factors, viral characteristics, and environmental exposures.

As summarized in Figure 1, key determinants of RSV-related asthma and wheezing include infection severity and age at first exposure, underscoring the need for early diagnostic tools and predictive markers to improve patient management.

The emergence of biomarker-based risk stratification, particularly through protein expression profiling of ILs and chemokines, represents a promising frontier in RSV research. These advances could enable personalized prevention and treatment strategies, ensuring that high-risk infants receive timely interventions to mitigate long-term complications. Looking ahead, future research should focus on validating and integrating biomarkers into clinical practice, optimizing early detection, and refining preventive strategies such as monoclonal antibody therapies and maternal vaccination. A multidisciplinary, proactive approach—combining biomarker-driven risk assessment, immunoprophylaxis, and tailored follow-up care—will be essential to reducing RSV-related morbidity and improving long-term respiratory health outcomes in pediatric populations. By advancing our understanding of RSV pathophysiology and leveraging innovative prevention and treatment strategies, we can significantly reduce the burden of RSV-associated respiratory diseases and enhance the quality of life for affected children.

Despite significant progress, several key questions remain unanswered. It is still unclear whether non-atopic children undergo a shift toward a Th2-skewed inflammatory profile following RSV infection, potentially predisposing them to asthma development. Additionally, further research is needed to identify which specific features of the preterm immune system most significantly contribute to increased susceptibility to severe RSV disease and subsequent asthma. Addressing these gaps through advanced methodologies such as single-cell analyses and multi-omics approaches will be critical for developing targeted interventions and improving long-term outcomes in vulnerable pediatric populations.

## Figures and Tables

**Figure 1 viruses-17-01073-f001:**
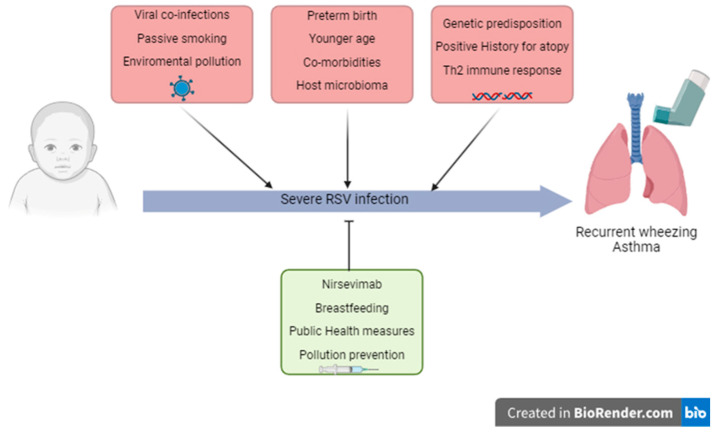
Risk and protective factors involved in severe RSV infection in childhood and the subsequent development of recurrent wheezing and asthma.

**Figure 2 viruses-17-01073-f002:**
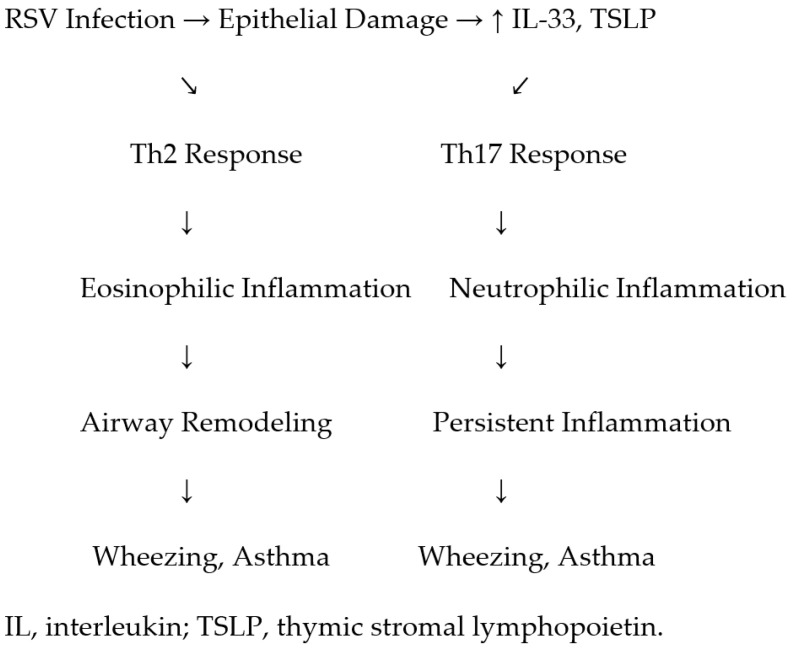
Proposed immunological pathways linking RSV infection to recurrent wheezing and asthma, highlighting Th2/Th17 imbalance, cytokine profiles, airway remodeling, and genetic susceptibility factors.

**Table 1 viruses-17-01073-t001:** Distribution of registered cases according to age (in years) and epidemic season, described as an absolute number and percentage over the total cases [n] recorded in the given season. The “POST COVID-19” column shows the total cases of season 2021–2022 to 2022–2023 and the “PRE-COVID-19” shows those of season 2014–2015 to season 2019–2020.

Age (Years)	Post COVID-19 (n/%) [n = 645]	20/21 (n/%) [n = 7]	PRE COVID-19 (n/%) [n = 610]
0–1	390 (60.4%)	5 (71.4%)	479 (78.5%)
1–2	106 (16.4%)	1 (14.2%)	78 (12.8%)
2–3	65 (10.1%)	1 (14.2%)	26 (4.3%)
3–4	48 (7.4%)	0 (0%)	14 (2.3%)
4–5	24 (3.7%)	0 (0%)	7 (1.1%)
5–6	12 (1.9%)	0 (0%)	6 (1%)

Adapted from Lodi et al. [4].

**Table 2 viruses-17-01073-t002:** Markers of severity and of recurrent wheezing/asthma after RSV bronchiolitis and biological samples to be dosed.

Marker	Sample	Associated with	Reference
IL-33	NPA	Severity	García-García et al., 2017 [73]
IL-6	NPA	Severity	Tabarani et al., 2013 [72]
IL-8	NPA	Severity	Oh et al., 2002 [40]
IFN-α	NPA	Severity	Vázquez et al., 2019 [71]
TSLP	Serum	Severity	Lee et al., 2012 [74]
MUC5AC	NPA	Severity	Feltes et al., 2003 [59]
LDH	NPA	Severity	Brown et al., 2015 [75]
Periostin	Serum	Recurrent wheezing/asthma	Bertrand et al., 2015 [76]
EDN	Serum	Recurrent wheezing/asthma	Kim et al., 2013 [77]
CD14	Serum	Recurrent wheezing/asthma	Soferman et al., 2004 [78]
IL-3	BALF/NPA	Recurrent wheezing/asthma	Bertrand et al., 2015 [76]
IL-12	BALF	Recurrent wheezing/asthma	Bertrand et al., 2015 [76]

LDH: lactate dehydrogenase; MUC5AC: Mucin 5AC; TSLP: thymic stromal lymphopoietin; NPA: nasopharyngeal aspirate; EDN: eosinophil derived neurotoxin; MIP-1α: macrophage inflammatory protein 1-alpha; BALF: bronchoalveolar lavage fluid.

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
