# Peer review of "Predictive Factors and Clinical Markers of Recurrent Wheezing and Asthma After RSV Infection"

_viruses, 2025, doi:10.3390/v17081073_

Round 1
Reviewer 1 Report
Comments and Suggestions for Authors
• Manuscript tackles a very important problem - the morbidity of RSV-infection in babies and infants and its effects on the risk of wheezing and asthma in later age. The authors provide very comprehensive and well-structured review of the risk factors, protective factors and the possible actions for treating and preventing the RSV-infections and their complications. The manuscript contains brief but very informative depiction of molecules participating the pathogenesis of RSV-induced airway inflammation and their possible role as biomarkers and objects of treatment.
• The manuscript provides thorough and multifaceted analysis of the problem.
• The manuscript provides very comprehensive analysis of the links between RSV infection and wheezing at genetic and metabolic level. I gives important information for further research and for assessment the possible biomarkers for disease severity and possible preventive measures..
• I find the methodology of the manuscript good, as well as the presentation of articles reviewed.
• The conclusions are consisted with the arguments presented, they address the main question.
• Reference is appropriate.
• Any additional comments on the tables and figures. Tables provided are very clear and informative.
Author Response
- Manuscript tackles a very important problem - the morbidity of RSV-infection in babies and infants and its effects on the risk of wheezing and asthma in later age. The authors provide very comprehensive and well-structured review of the risk factors, protective factors and the possible actions for treating and preventing the RSV-infections and their complications. The manuscript contains brief but very informative depiction of molecules participating the pathogenesis of RSV-induced airway inflammation and their possible role as biomarkers and objects of treatment.
Re: Thank you very much for your positive evaluation. We revised the manuscript according to the comments received from the other reviewers.
- The manuscript provides thorough and multifaceted analysis of the problem.
Re: Thank you very much for your positive evaluation.
- The manuscript provides very comprehensive analysis of the links between RSV infection and wheezing at genetic and metabolic level. I gives important information for further research and for assessment the possible biomarkers for disease severity and possible preventive measures..
Re: Thank you very much for your positive evaluation.
- I find the methodology of the manuscript good, as well as the presentation of articles reviewed.
Re: Thank you very much for your positive evaluation.
- The conclusions are consisted with the arguments presented, they address the main question.
Re: Thank you very much for your positive evaluation.
- Reference is appropriate.
Re: Thank you very much for your positive evaluation.
- Any additional comments on the tables and figures. Tables provided are very clear and informative.
Re: Thank you very much for your positive evaluation.
Reviewer 2 Report
Comments and Suggestions for Authors
This review systematically summarizes the major clinical predictors and biomarkers associated with the development of recurrent wheezing and asthma in children following RSV infection. It emphasizes the multifactorial mechanisms involving infection severity, age, prematurity, immune dysregulation, and genetic susceptibility.
1. The logical flow between sections is weak. For example, Section 3, "Recurrent wheezing and asthma risk factors," combines various dimensions such as immunity, microbiota, and genetics without clear internal categorization or subheadings to guide the reader.
2. The review needs to incorporate recent findings from real-world evidence following the introduction of nirsevimab (2022–2024), especially from large prospective biomarker cohorts such as COPSAC and INSPIRE.
3. The discussion on immune mechanisms—such as Th2/Th17 imbalance, IL-33, and TSLP—should be supported by an integrated mechanistic diagram or pathway illustration. The mention of multiple SNPs in the genetic susceptibility section also requires deeper discussion in the context of GWAS and single-cell analysis.
4. The distinctions between recurrent wheezing and asthma in terms of diagnostic criteria, pathological mechanisms, and prognosis should be further clarified.
Author Response
This review systematically summarizes the major clinical predictors and biomarkers associated with the development of recurrent wheezing and asthma in children following RSV infection. It emphasizes the multifactorial mechanisms involving infection severity, age, prematurity, immune dysregulation, and genetic susceptibility.
Re: Thank you very much for your comments. We revised the manuscript according to your suggestions and those received from another reviewer.
- The logical flow between sections is weak. For example, Section 3, "Recurrent wheezing and asthma risk factors," combines various dimensions such as immunity, microbiota, and genetics without clear internal categorization or subheadings to guide the reader.
Re: Section 3 has been revised according to your comments (pp. 3-10).
- The review needs to incorporate recent findings from real-world evidence following the introduction of nirsevimab (2022–2024), especially from large prospective biomarker cohorts such as COPSAC and INSPIRE.
Re: Recent findings have been incorporated as suggested (pp. 10-11).
- The discussion on immune mechanisms—such as Th2/Th17 imbalance, IL-33, and TSLP—should be supported by an integrated mechanistic diagram or pathway illustration. The mention of multiple SNPs in the genetic susceptibility section also requires deeper discussion in the context of GWAS and single-cell analysis.
Re: A Figure has been added as recommended (pp. 7-8). Moreover, the text has been revised as suggested (p. 9).
- The distinctions between recurrent wheezing and asthma in terms of diagnostic criteria, pathological mechanisms, and prognosis should be further clarified.
Re: Clarified as suggested (p. 6).
Reviewer 3 Report
Comments and Suggestions for Authors
Buttarelli L et al., presented an interesting narrative review on predictive values and clinical markers of recurrent wheezing and asthma after RSV infection. They summarized literature data pointing out that predictive factors associated with recurrent wheezing and asthma include severity of the initial infection, age at exposure, genetic susceptibility, prematurity, air pollution, and tobacco smoke. A description of biomarkers which could be involved in the occurrence of repeated wheezing and asthma development were presented. The review is particularly interesting and well presented. I have only a few comments which could better define some aspects.
COMMENTS:
- Authors hypothesized different mechanisms of response to RSV in term and pre term children however, the only study they presented with a comparison between these two populations after RSV demonstrated equal amount of cytokines in nasal swabs. Hypothesis are very likely but are any other studies comparing the inflammatory/immune response in term and pre term children infected with RSV?
- The paragraph regarding the role of environmental factors should be implemented. It would be better to cite original articles than only one review on the matter.
- In figure 1, all the risk factors are referred to the development of severe infection while no risk factors are indicated for the development of recurrent wheezing and asthma, apart from severity. In the text the risk factors for developing recurrent wheezing and asthma were: severity of infection, need for hospitalization, longer hospital stays, and the necessity for intensive care treatment. Risk and predictive factors both for severity and for recurrent wheezing and asthma should be better distinguished
- Table 2 dos not report IL-8 and IFNa among markers of severity. Each biomarker should be in a separate raw and a column and specific references should be added. In general, I found a bit of confusion between biomarkers possibly involved both in severity and in the occurrence of repeated wheezing and asthma and those really demonstrated in follow-up studies. Authors should better distinguish possible and demonstrated mechanisms.
- In the paragraph “Markers of RSV infection” the review by Vázquez Y 2019 was often cited. It would be better to directly cite the original studies reported in that review.
- Are there any preliminary data evaluating the immune profile of children treated with Nirsevimab who still had severe RSV infection compared to those protected from RSV infection?
- In the paragraph 3.4, I don’t clearly understand whether atopic patients who experienced RSV infections had more probability to develop recurrent wheezing and asthma than atopic subject without RSV infection.
- There are some points which probably were not clearly addressed by published studies and that could be included at the end of the discussion as prospective studies: Do non atopic patients change their inflammatory profile towards a Th2 profile after a RSV infection? Which characteristic(s) of pre-term immune system could have a major impact in determining severity of infection and asthma development?
- Line 207: please include the duration of the follow-up
Author Response
Buttarelli L et al., presented an interesting narrative review on predictive values and clinical markers of recurrent wheezing and asthma after RSV infection. They summarized literature data pointing out that predictive factors associated with recurrent wheezing and asthma include severity of the initial infection, age at exposure, genetic susceptibility, prematurity, air pollution, and tobacco smoke. A description of biomarkers which could be involved in the occurrence of repeated wheezing and asthma development were presented. The review is particularly interesting and well presented. I have only a few comments which could better define some aspects.
Re: Thank you for your comments. We revised the manuscript according to your suggestions and those received from another reviewer.
COMMENTS:
- Authors hypothesized different mechanisms of response to RSV in term and pre term children however, the only study they presented with a comparison between these two populations after RSV demonstrated equal amount of cytokines in nasal swabs. Hypothesis are very likely but are any other studies comparing the inflammatory/immune response in term and pre term children infected with RSV?
Re: Your suggestion has been included in the text (pp. 8-9).
- The paragraph regarding the role of environmental factors should be implemented. It would be better to cite original articles than only one review on the matter.
Re: The paragraph on environmental factors has been re-written and expanded (pp. 11-12).
- In figure 1, all the risk factors are referred to the development of severe infection while no risk factors are indicated for the development of recurrent wheezing and asthma, apart from severity. In the text the risk factors for developing recurrent wheezing and asthma were: severity of infection, need for hospitalization, longer hospital stays, and the necessity for intensive care treatment. Risk and predictive factors both for severity and for recurrent wheezing and asthma should be better distinguished.
Re: A paragraph on recurrent wheezing and asthma has been included (p. 4).
- Table 2 dos not report IL-8 and IFNa among markers of severity. Each biomarker should be in a separate raw and a column and specific references should be added. In general, I found a bit of confusion between biomarkers possibly involved both in severity and in the occurrence of repeated wheezing and asthma and those really demonstrated in follow-up studies. Authors should better distinguish possible and demonstrated mechanisms.
Re: Table 2 has been replaced by a new one (pp. 14-15).
- In the paragraph “Markers of RSV infection” the review by Vázquez Y 2019 was often cited. It would be better to directly cite the original studies reported in that review.
Re: Original studies have been included (pp. 13-17).
- Are there any preliminary data evaluating the immune profile of children treated with Nirsevimab who still had severe RSV infection compared to those protected from RSV infection?
Re: Real-word data and a comment on immune profile after nirsevimab administration have been added (pp. 12-13).
- In the paragraph 3.4, I don’t clearly understand whether atopic patients who experienced RSV infections had more probability to develop recurrent wheezing and asthma than atopic subject without RSV infection.
Re: A specific paragraph has been added (p. 9).
- There are some points which probably were not clearly addressed by published studies and that could be included at the end of the discussion as prospective studies: Do non atopic patients change their inflammatory profile towards a Th2 profile after a RSV infection? Which characteristic(s) of pre-term immune system could have a major impact in determining severity of infection and asthma development?
Re: Added as requested (pp. 18-19).
- Line 207: please include the duration of the follow-up
Re: Added as recommended (p. 5).
Round 2
Reviewer 2 Report
Comments and Suggestions for Authors
none